# Biophysical Techniques for Target Validation and Drug Discovery in Transcription-Targeted Therapy

**DOI:** 10.3390/ijms21072301

**Published:** 2020-03-26

**Authors:** Mehdi Moustaqil, Yann Gambin, Emma Sierecki

**Affiliations:** EMBL Australia Node in Single Molecule Science and School of Medical Sciences, UNSW Sydney, NSW 2052, Australia; y.gambin@unsw.edu.au

**Keywords:** transcription factors, biophysical techniques, drug discovery, therapy

## Abstract

In the post-genome era, pathologies become associated with specific gene expression profiles and defined molecular lesions can be identified. The traditional therapeutic strategy is to block the identified aberrant biochemical activity. However, an attractive alternative could aim at antagonizing key transcriptional events underlying the pathogenesis, thereby blocking the consequences of a disorder, irrespective of the original biochemical nature. This approach, called transcription therapy, is now rendered possible by major advances in biophysical technologies. In the last two decades, techniques have evolved to become key components of drug discovery platforms, within pharmaceutical companies as well as academic laboratories. This review outlines the current biophysical strategies for transcription manipulation and provides examples of successful applications. It also provides insights into the future development of biophysical methods in drug discovery and personalized medicine.

## 1. Introduction

Converting genetic information into messenger RNA for subsequent protein translation is a fundamental event for every cell. Transcription is tightly regulated by a large number of proteins (e.g., transcription factors and transcription co-regulators) that bind to DNA, specifically at cis-acting elements residing in the 5′ promoter region. Since fine control of transcription is essential, it is not surprising that dysregulation in the assembly of the transcriptional machinery is often linked to the pathophysiology of human disease [1]. 

In the last decade, studies have shown that mutations in both coding transcription factors (TFs) and non-coding TFs (such as those maintaining regulatory DNA elements) can be the root causes of human diseases. The idea that the disruption of specific protein interactions can lead to human disease [2] has come to complement the official gene loss/perturbation concept [3]. Therefore, therapeutic intervention to rewire those networks is becoming an exciting new avenue for drug development.

Transcription therapy is a new approach that aims at rectifying aberrant gene expression through direct pharmacological intervention on the transcription process. This term was only coined in 2001 [4], yet the underlying principle has, willingly or by accident, existed for longer. It is estimated that at least 10% of FDA-approved anti-cancer drugs modulate transcription [5], even though that was not their intended mode of action. Transcription is often considered “undruggable” as it is a nuclear event not readily accessed by therapeutic agents. Further, key players in transcription often lack the enzymatic activity that can be readily targeted by chemical intervention, contrary to many proteins involved in cell signaling (e.g., protein kinases). But the view is evolving and some transcription-targeted therapeutic agents have entered clinical trials or been approved for clinical practice, for cancer treatments [6].

## 2. Targeting Transcription Factors: Transcription Therapy

TF dysregulation is fundamental to the pathogenesis of many types of human disease. It is evident in, but not limited to congenital disorders and, for example, TFs greatly contribute to tumorigenesis and malignancy. TFs are thereby drug target candidates with great potential for disease therapy. Currently, there are very few approved drugs that directly target TFs. Even molecular probes that interfere with TF function in animal models are scarce. This is because TFs operate in an elusive manner that is typically very different from traditional drug targets, leading to the premature assessment that TFs are undruggable. Indeed, the vast majority of marketed drugs today, for cancer therapy or any other disease, act through the inhibition of enzymatic activity or ligand binding—two key features that are notoriously absent in TFs (with the exception of nuclear receptors). To put this is the context of cancer, only about half of oncogenes can be targeted in this manner, meaning that a very large portion of potential drug targets is instantaneously lost through the exclusion of TFs [7]. Add to this that many non-oncogene TFs are also drivers of tumour growth and malignancy, and therefore potential molecular targets for cancer therapy.

Targeting multiple facets of tumour growth with combination treatment is a rational strategy but requires the administration of multiple pharmaceuticals and therefore has its limitations. Yet, the multiple proteins that are targeted in this scenario are unequivocally the products of genes regulated by TFs. Similarly, downstream of signalling pathways is a transcriptional response that is also controlled by TFs. Therefore, modulation of these TFs at the point where cell signalling converges, is likely to be much more effective than targeting individual protein players in a highly interconnected signalling pathway. 

Genetic approaches in animal models, as well as initial pharmacological approaches, demonstrate the potential of manipulating TF function in vivo [8,9,10]. It needs to be said though, that the therapeutic targeting of a single TF does not ensure its complete loss of function, or the disruption of the molecular process under its control. Drug resistance can also occur in the case of TFs, as is demonstrated in the case of the estrogen receptor (ER). ER is a nuclear receptor that is activated by an endogenous ligand (estrogen) and is prominently involved in human breast cancers [11,12]. Drugs preventing ligand binding and subsequent activation of ER have been developed and used as anti-cancer agents. Resistance to ER-based therapies can arise, caused either by epigenetic changes or subtle mutations to the *ER* coding sequence, leading to the apparition of ligand-independent ER activity [12,13,14,15]. Although the drug resistance of the ER is associated with its ligand binding domain—which sets nuclear receptors apart from most TFs—other TFs may also find loopholes to thwart the long-term efficacy of TF-targeted therapies. Hence, it is important to understand the molecular mode of action of a TF, particularly how it achieves activity and selectivity, and to appreciate its individual place in driving a biological (and pathogenic) process.

### 2.1. Strategies to Target Transcription Factors

There are multiple ways in which we can interfere with the functionality of TFs, including altering the absolute abundance of a given TF, either by regulating how much of the protein is being produced or by regulating proteolytic degradation. Another approach is to alter the relative abundance of TFs in the nucleus (where a TF is active) by modulating post-translational modifications, such as sumoylation and phosphorylation [16,17,18], that affect nuclear shuttling. However, these strategies do not physically target TFs *per se* and are therefore subject to the limitation of drugging conventional enzyme targets in upstream cell signalling. Hence, to take full advantage of therapeutically targeting TF at the point of convergence in cell signalling, drugs should interfere with the capacity of TFs to regulate transcription, leading to the disruption of a key biological output such as cell type specific proliferation or differentiation. 

When considering TFs as potential therapeutic targets, we generally assume that the potential lies in antagonists that inhibit pathogenic hyperactivity, for instance in the case of oncogenes. However, a great potential also lies in the development of agonists that can constitutively activate a TF, as activation of tumour suppressor genes, for example, could be beneficial in cancer therapy. 

### 2.2. Transcription: A Complex Process That Can Provide Multiple Targets

During transcription, the transcription machinery dynamically regulates the copy of genetic information stored in DNA into units of transportable complementary RNA. Transcription is a complex process involving multiple stages. Through focussing on TFs, it can be pharmaceutically targeted at least three distinct levels [19] (Figure 1).

#### 2.2.1. Chromatin Remodelling and Epigenetics

The first level of regulation is related to the modification of the epigenetic landscape, including promoter methylation and posttranslational modifications of core histones. This step is crucial as only the euchromatin (loose or open chromatin) structure is permissible for transcription, while heterochromatin (tight or closed chromatin) is more compact and refractory to binding of factors, such as TFs, that need to gain access to the DNA template. Epigenetic regulators control protein function and stability, and impact gene transcription, DNA replication and DNA repair. They produce potentially heritable changes in gene function without modifying the underlying DNA and so should be at the forefront of novel strategies to disrupt TF activity. The fact that epigenetic alterations are often observed in human cancers [20] make therapeutics targeting epigenetic modifications promising anti-cancer candidates. These therapeutic agents often target histone deacetylases, as well as other proteins that have an intrinsic enzymatic activity, making them druggable in a traditional way. Clinical trials have commenced on drugs targeting these regulators, such as enhancer of zeste homologue 2 (EZH2), disruptor of telomeric silencing 1-like (DOT1L) and arginine methyltransferase 5 (PRMT5) protein.

#### 2.2.2. Recruitment of TFs to Cis-regulatory Elements

The second level of control consists in preventing binding of TFs to defined promoter/enhancer regions of the chromatin. Preventing a TF from binding to the regulatory sequences on the DNA is indeed the simplest way to interfere with the activity of a TF. This can be achieved by targeting the DNA-binding domain (DBD) of the target TF or mimicking cis-regulatory elements to create “protein traps”. An alternative approach is to bind directly to the DNA, effectively masking the DNA regulatory element. 

Inhibitors targeting the DBD have been successful in disrupting TF-DNA interactions, as demonstrated in vitro for B-ZIP TFs (e.g., CREB) [21], STAT3 [22] and SOX2 [23]. Clinical trials are also underway for their use in humans, however, with mixed outcomes [24,25]. DBD are highly conserved between members of TF families [26] and thus selective inhibition is challenging [27]. Further, due to the interconnected gene regulatory networks (GRNs) that control gene expression, a therapy based on preventing TF-DNA binding is sensitive to rescue mechanisms by redundant TFs. This is exemplified by the fact that dominant negative mutations in TFs result in much more severe phenotypes than loss of function mutations [28].

This effect can be overcome by exploiting the specificity of the DNA recognition motif. Because the TF-DNA interface is a high affinity interaction in the nM to sub-µM range [29,30,31], complex macromolecules, such as oligonucleotide decoys or polyamides, are required to compete with these interactions. Development and production of such molecules can be costly. Nevertheless, oligonucleotides decoys have been developed for the inhibition of STAT3 [22,24]. These have however poor pharmacokinetic properties (bioavailability and half-life), limiting their use.

Highly selective polyamides can mimic the protein and prevent binding of a TF to a specific DNA sequence, and pyrrole–imidazole polyamide minor groove binders have been used for the inhibition of NF-κB [32]. The increased selectivity is potentially a double-edged sword though. Indeed, a TF often recognizes more than one specific motif when it comes to gene regulation. Specific DNA-binders will only interfere with a subset of a TF target genes, and only affect the corresponding biological processes [33]. This was suggested to potentially enhance specificity of treatment [33], which has merit since specific enhancers can be associated with the function of a TF in a specific cell type. Ultimately though, it the efficacy of disease therapy in complex pathologies is likely to be limited if only a fraction of the TF activity is affected. In addition, a molecule binding directly to DNA is likely to cause topological changes in the genome landscape, particularly if targeting pioneer TF target sites, potentially causing unpredictable secondary effects. 

#### 2.2.3. Targeting Protein Complexes

The third level of regulation is achieved through modulating protein–protein interactions (PPIs). These can occur between TFs and their regulatory proteins (often referred to as transcriptional co-regulators), or between TFs themselves, including homodimerization. As transcription is a concerted mechanism, therapeutic agents disrupting PPIs result in altered gene expression. The formation of protein complexes is indeed absolutely critical to the functionality of TFs. Often TFs in isolation don’t have enough binding affinity to associate with DNA regulatory elements (with the exception of pioneer TFs). Interactions with protein partners are fundamental to the conformation of a TF, including the DNA-binding domains (DBD) that are often highly flexible. Furthermore, many of the protein partners of a TF are context-dependent (e.g., cell type specific) and will dictate the selectivity towards target genes. As illustrated in Figure 2, depending on the context, PPI disruption can either have a targeted effect or a broad impact, when a network hub is affected. Therefore, interference with TF-protein partner interactions has high intrinsic value for disease therapy. In fact, interference with PPIs is valued to such an extent that conventional non-TF drug targets are also increasingly targeted at the level of protein interactions—despite the challenges [34,35,36,37,38].

For more than 20 years, targeting PPIs has been a dream in the drug discovery field. Protein complexes are ideal drug targets. The majority of marketed drugs target less than 300 of the ~20,000 proteins encoded in the human genome, as only a minority of proteins have suitable binding sites for small molecules to modulate their activity. As previously mentioned, enzymes are the main target group for drug development. Typically, the binding sites are highly conserved between protein families due to a restrictive enzyme catalytic site, which compromises drugs specificity. In contrast, the idea of targeting the complex and unique interfaces that characterize PPIs significantly expands the landscape of drug targets, and opens the possibility of finding highly specific, high-affinity binders. The main hurdle in disrupting PPIs is that the interfaces between proteins, as in the case of TFs and protein partners, are seen as very large [39], typically 500–2000 Å^2^. In average, a 10 Å long small molecule can bind to deep, well-defined hydrophobic cavities of < 500 Å^2^ [40]. This suggests that larger molecules, bearing multiple aromatic moieties are required [35,37]. Peptide-based PPI disruptors (e.g., stapled peptides) have also been successfully developed, with high affinity and selectivity in vitro, but they have not progressed further than pre-clinical disease models [41,42]. Peptidomimetics can offer an alternative, though the increasing sizes of these drug-like molecules can adversely affect bioavailability and cell membrane permeability [43]. The subsequent translation from homogenous in vitro binding assays into animal models or patients may be difficult and can require sophisticated compound stabilization and delivery methods. In the recent years, the realization that the binding energy of an interaction is not distributed evenly over the entire length of an interface has further promoted the development of small molecule PPI disrupters. Indeed, small interaction hot spots around the centre of the PPI interface confer most of the binding energy [35,44], while making up less than half of the surface. This generates more concentrated binding pockets of 250–900 Å^2^ that are generally more hydrophobic than the rest of the interface, and much more suitable for the binding of small molecule inhibitors [45,46]. The paradigm is, therefore, changing and it is now a research focus to disrupt PPIs by targeting localized hotspots at the surfaces of these interactions [45,47].

Overall, in the last three years, the number of small molecule inhibitors that target PPIs has increased by 5-fold to 242 compounds (not exclusively for TFs), targeting 26 different PPIs [45,48]. The chemical space of PPIs and their small molecule inhibitors has been recorded in the online databases such as the PPI inhibition database (2P2IDB, http://2p2idb.cnrs-mrs.fr/) or more generally in the RCSB protein database (PDB, www.rcsb.org) [49], which provides a good insight into the 3D PPI surface, interacting residues and orientation of small molecule inhibitors. 

Typical examples of PPI disruptors targeting TFs include inhibitors of the MYC/MAX heterodimer. MYC is overexpressed in many different types of cancers, and its endogenous and pathogenic function is completely dependent on the interaction with its partner protein MAX [50], as only the heterodimer can bind DNA. Peptidomimetics [51,52,53] that disrupt MYC/MAX dimerization have shown efficacy in in vivo cancer model systems [54].

The main limitation of this strategy, in the case of TFs, is the determination of the therapeutic targets, as little is known about how TFs function co-operatively. Over the past decade however, several high-throughput methodologies have been elaborated and have led to the standardized mapping of interactions between TFs [55]. The first generation of these tools include computational reverse engineering [56,57], chromatin immunoprecipitation (ChIP) combined with microarrays (ChIP-chip), next generation sequencing (ChIP-seq) [58,59] or proteomics (RIME) [60]. Biophysical techniques have reinforced and validated the database of PPIs underlying the TFs GRNs [61,62].

In this review, we illustrate how the use of new biophysical techniques can facilitate the discovery of novel therapeutic targets that target the transcriptional activity of TFs by disrupting protein-protein interactions. 

## 3. Techniques for Target Validation and Drug Screening

Since the first successes with structure-based drug design using X-ray crystallography in the 1990s [63], pharmaceutical companies and academic laboratories have expanded their drug discovery platforms by acquiring and developing a wide range of complementary biophysical technologies. Here we will outline the biophysical techniques that have or can be utilized for target validation and drug discovery in transcription therapy. These techniques are illustrated in Figure 3. Given the large number of biophysical techniques, this review is inevitably not able to comprehensively cover all existing or developing techniques. Further details on the mentioned techniques or description of other (unmentioned) techniques can be found elsewhere.

### 3.1. Generalist and Structural Techniques

There is a multitude of extensively used and novel techniques employed in vitro and in vivo to uncover the 3-dimensional structure and other general information, from protein sequence and mutation identification to the dynamic structural changes as a function of pH or interactions with other proteins.

#### 3.1.1. Mass Spectrometry (MS)

Mass spectrometry (MS) has become a reference tool in drug target discovery and is providing valuable answers in many areas of biomedical research. MS from intact tissues transformed molecular biology and biochemistry by providing sensitive, rapid and specific analyses of peptides and proteins. Indeed, MS provides information on molecular weights with high mass accuracy, permitting the identification of proteins from peptide sequencing combined with protein database searches. It can also detect protein modifications such as phosphorylation and acetylation or analyze protein complexes. Tissue profiling and imaging by mass spectrometry (MSI) is at an early stage of development but promises to expand our understanding of normal biological and pathological processes. This tool can interrogate protein expression in tissues, in a high throughput manner. Although improvements in sample preparation protocols, instrumentation, and data analysis are still required, the application of MSI to several clinical and biological problems exemplifies the fundamental contributions of this technology. MSI provides molecular weight-specific maps (or MSI images), at relatively high resolution and sensitivity. These images have proved useful in investigating pathologies, monitoring effect of chemotherapeutics, and discovering new disease biomarkers. 

One of the existing applications of MALDI MS is to measure the predisposition for and response to external agents by tumors and surrounding tissues. First, the selection of therapeutic agent can be influenced by the original protein profile obtained from the primary tumor. The level of delivery of a drug to a particular site can be measured directly, from a tissue biopsy. Measuring the ability of drugs and other bioreactive molecules to effectively penetrate larger tumors is indeed problematic and could be boosted by this technology. In addition, by comparing results immediately after introduction of a therapy with controls at regular intervals, alterations in specific molecular pathways, directly or indirectly modulated by the agent, can be monitored. Many studies of this type clearly establish proof of principle [64] for the field. Furthermore, similar methods are also envisioned to monitor the remission in patients treated with traditional therapy. The information obtained from tissue profiling and MSI dramatically expands but is not exclusive of existing molecular biological techniques. Rather, these complementary tools will promote a better understanding and assist new discoveries in biology and medicine.

#### 3.1.2. X-ray Crystallography

X-ray crystallography is part of the main biophysical techniques currently used to investigate the structure of protein–ligand and protein-protein complexes. X-ray crystallography can be used for proteins of any size. It is the most powerful, robust and routine method for providing a detailed atomic picture of a compound binding to its target. X-ray diffraction patterns of either protein–ligand co-crystals or apoprotein crystals soaked with ligands give atomic details of the structure of the protein–ligand complex [64]. The main limitation is the need to obtain crystals, which usually requires that at least a few milligrams of protein, at more than 10 mg/mL, be obtained. This can be a significant and sometimes unsurmountable obstacle when proteins are difficult to obtain, as is often the case with TFs. Very recent developments aim at eliminating the crystallization stage. X-ray crystallography can now be performed on crystal slurries and cryo-EM could attain atomic resolution on micro-crystals (see later). These very recent technologies have not entered the drug development world yet but will have a huge impact on target identification in the future.

Another way to bypass the need of crystals is by using Small-angle X-ray scattering (SAXS). SAXS has been used to study protein–protein, protein-DNA, protein-RNA and protein-small molecule interactions in solution. It provides information on the folding, oligomerization state and intrinsic flexibility of a protein and its complexes, as well as the shape of the assembly in an envelope structure with a 1–2-nm resolution [65,66]. In the future, SAXS may emerge as a routine tool in studies of weak protein-ligand interactions. On modern synchrotrons, X-ray experiments can now be performed in millisecond time frames, giving access to kinetic processes, as is the case in time resolved SAXS studies. 

#### 3.1.3. Nuclear Magnetic Resonance (NMR)

NMR spectroscopy is known for its ability to characterize macromolecular structures, as well as molecular and supramolecular dynamics. It is therefore the preferred tool to investigate both static and transient features of proteins. NMR is also a valuable screening tool recording binding of ligands to protein targets. A key advantage of NMR is the ability to detect and quantify even transient interactions, with high sensitivity and without prior knowledge of the protein structure and function. NMR is particularly useful when working with TFs as it can assess dynamic structures. Indeed, the prevalence of intrinsically disordered regions in TFs means that these proteins do not maintain a rigid 3D structure, rather they change conformation as a function of their environment, a feature that cannot be detected by X-ray. Furthermore, NMR provides structural information on both the target and the ligand to aid subsequent optimization of weak-binding hits into high-affinity leads. NMR achieves such specificity through two alternative experimental setups: ligand-observed NMR and protein-observed NMR. Both methods require relatively large amounts of protein, which needs to be stable and of high purity. In ligand-observed NMR, changes in the NMR parameters (mostly chemical shifts, relaxation rates and diffusion rates) of molecules or cocktails of molecules are measured in the presence of a target protein. Typical experiments performed include magnetization transfer experiments (saturation transfer difference (STD) NMR, water-ligand observed via gradient spectroscopy (waterLOGSY), transferred nuclear Overhauser effect (NOE), NOE pumping and other NOE-based methods), relaxation editing (longitudinal, transverse and double-quantum relaxation) and diffusion editing. 

In vivo experiments present a new challenge for NMR technique and the extrapolation of these ligand-observation-based strategies will soon allow us to study binding in living cells [67]. We anticipate with much excitement the continued symbiosis of NMR techniques and drug design, as well as the interdisciplinary collaboration of all structural analysis techniques for drug target identification especially directed at TFs.

### 3.2. Protein-Protein Interactions Interrogation Techniques

These biophysical techniques are used to study the properties, dynamics and function of protein complexes at the molecular level.

#### 3.2.1. Surface Plasmon Resonance (SPR)

Optical biosensors, exploiting mostly surface plasmon resonance (SPR), have been extensively used in drug discovery for two decades, for compound screening and lead optimization [68]. SPR is a spectroscopic technique that relies on changes in refractive index at the interface between a liquid sample and the surface on which a sensor, typically the target protein, is immobilized [69]. Upon analyte binding or inducing conformational changes, the signal is shifted. Continuous registration of the signal, using microfluidic systems, gives access to the binding mechanism and corresponding kinetic parameters: the association rate constant (k_on_), dissociation rate constant (k_off_) and resulting affinity (K_d_). Future progress in SPR probably lies in SPR imaging and microscopy, following the current exploratory array-based developments (with nanohole arrays using diffraction, nanowires, or nanorods) [65] that would ultimately facilitate the monitoring of any kind of molecular interaction and allow SPR biosensors to be used in every aspect of transcriptional therapy, from target identification and characterisation, to supporting clinical trials and the production of the novel drugs.

#### 3.2.2. Isothermal Titration Calorimetry (ITC)

Isothermal titration calorimetry (ITC) is established as one of the preferred approaches to study bimolecular interactions. It is mostly used in the study of macromolecule-macromolecule or macromolecule-small molecule interactions [70]. The main advantages of calorimetric methods in general, and ITC in particular, is to bring invaluable information for drug design [71] that cannot be obtained by other means. Indeed, a key step in drug development is to obtain inhibitors and ligands with high binding affinities for their target molecules. However, the precise determination of the binding affinity becomes increasingly more difficult as affinity approaches and surpasses the nanomolar level. Specific protocols in ITC, such as an experimental mode called displacement titration [70], can measure interactions down to the picomolar range giving access to the complete binding thermodynamics of a ligand. The downside so far is that measurements are typically time-consuming and not yet available in a high-throughput format, but this is changing.

#### 3.2.3. Microscale Thermophoresis (MST)

Microscale thermophoresis (MST) is a relatively new methodology that monitors fluorescence in an infrared laser-heated spot. It is an equilibrium-based method that can detect ligand binding-induced changes in thermophoretic mobility (the motion of molecules along a microscopic temperature gradient). Thermophoretic mobility varies depending on size, charge and hydration shell. These changes can be used to estimate K_d_ values [72]. Although MST is increasingly used to assess both protein-protein and protein-ligand interactions, the theoretical framework of the assay renders it difficult to predict the resulting signal, potentially masking interesting mechanistic details.

#### 3.2.4. Affinity Chromatography

Affinity chromatography exploits the differences in interactions’ strengths of the different biomolecules between a stationary and a mobile phase. The stationary phase containing a variety of biomolecules (DNA or proteins, depending on the purification experiment) is first loaded into a column with mobile phase. Then, the two phases are allowed time to bind. A wash buffer, poured through a column, removes non-target biomolecules by disrupting their weaker interactions with the stationary phase. Target biomolecules remain bound to the stationary phase due to a much higher affinity for the stationary phase. The elution buffer is used to disrupt the remaining interactions, effectively removing the target biomolecules [73]. 

This technique efficiently isolates proteins by taking advantage of their affinities for specific molecules—including substrates, inhibitors, antigens, ligands, antibodies or subunits of a target complex. It is a powerful approach as, even if the properties of a protein are unknown, affinity chromatography can be applied to identify interactions. 

#### 3.2.5. Immunoprecipitation

Immunoprecipitation (also called pull-down assay) consists in isolating an antigen by binding to a specific antibody, attached to a sedimentable matrix. The source of the antigen can be diverse: unlabeled cells or tissues, metabolically or extrinsically labeled cells, subcellular fractions from either unlabeled or labeled cells, or recombinantly expressed proteins [74]. The combination of immune-precipitation and mass spectroscopy is now commonly used to define the interactome of a target protein.

#### 3.2.6. ELISA

Enzyme-linked immunosorbent assay (ELISA) is the most commonly used method in analysing biomolecules. As a simple, rapid and specific assay, ELISA has been used as a research tool as well as a widely adopted diagnostic method in clinical settings and for microbial testing in all types of laboratories. Bimolecular interactions such as PPIs can be easily studied with this technique. Inhibition ELISA is a one-site binding analysis method which can monitor PPIs in solution and an improvement compared to more commonly used sandwich ELISA in which capture of the analyte on a solid surface is required, either through specific capture or through passive adsorption.

#### 3.2.7. Alpha Screen

Amplified Luminescent Proximity Homogeneous Assay Screen (AlphaScreen) is a versatile assay technology developed to measuring analytes using a homogenous protocol. It is bead-based proximity assay and was developed from an initial diagnostic assay technology known as Luminescent Oxygen Channelling Assay (LOCI) [75]. In brief, singlet oxygens, generated by high energy irradiation of “donor” beads, can only travel over a specific distance (approx. 200 nm) to “acceptor” beads. Inside “acceptor” beads, a cascading series of chemical reactions leads to the generation of a chemiluminescent signal. AlphaScreen has been widely deployed for cell signalling research and biomarker quantification [61,76], as well as drug discovery, principally high throughput screening (HTS) [77]. This wide adoption results from both the simplicity of the protocols and the high sensitivity of the assay. These, along with the fact that wash and separation are not necessary, allowed assays to be designed for automated liquid handling and detection instrumentation frequently used in HTS [78]. 

#### 3.2.8. Förster Resonance Energy Transfer (FRET)

Following photoexcitation, energy absorbed by a fluorescent molecule can be transferred efficiently over a distance of up to several tens of Angstroms to another fluorophore by the process of resonance energy transfer (RET). Förster resonance energy transfer (FRET), which involves the nonradiative transfer of excitation energy from an excited donor to a proximal ground-state acceptor, is a well-characterized photophysical tool [79]. It is very sensitive to nanometer-scale changes in donor–acceptor separation distance as well as to their relative dipole orientations. It has found a wide range of applications in analytical chemistry, protein conformation studies, and biological assays. In the last decade, FRET became a widely used fluorescence-based technique due to its potential advantages for studying the biological processes in living cells and its incredibly low noise-signal ratio. Intramolecular FRET investigations have revealed switches in conformational structures of proteins [80]. 

### 3.3. In Cellulo Techniques

While the previous techniques are mainly used in vitro, it is crucial to monitor and study PPIs in cellular contexts as well.

#### 3.3.1. Imaging

Cellular imaging has regained interest in the last decade with the development and generalization of microscopes with better resolution and faster acquisition. The generalized use of confocal microscopy to study not only the cell architecture but also the cellular localization of proteins has provided valuable biological information. In the case of TFs, cellular imaging can be useful to follow the dynamics of nuclear shuffling, for example. Very recent developments in single-molecule imaging and high-content imaging techniques are still at the early experimental stages but promise to deliver a wealth of information on the mechanisms of transcription (see Perspectives).

Yet, cellular imaging can be useful even when it is not providing molecular information. Advances in high-content imaging and analysis in high-throughput format have put multiplexed cellular content imaging to the forefront of drug discovery methods. These techniques provide a broader view of a molecule’s effect, with easier translation into animal models, progressively shifting the pendulum from molecular target-based strategies back to phenotypic screening. Overall, combination of high-content imaging and high-throughput screening systems have allowed the evaluation of tens or hundreds of thousands of compounds/drugs, and the narrowing down of potential candidates, with the use of automated machines to dispense cells and drugs, and to execute endpoint assays [81]. In silico methods and the development of artificial intelligence for big data analysis have also become important in drug discovery and drug repositioning [82]. 

#### 3.3.2. Protein-Fragment Complementation Assay (PCA)

Protein-fragment Complementation Assays (PCA)-based strategies provide a general methodology to detect and study spatial and temporal dynamics of PPIs. This methodology can be used with or instead of traditional target-based drug discovery strategies [83]. 

These assays make use of split reporter proteins to inform on the interaction/proximity between two proteins of interest [84]. The principle of PCAs is as follows: proteins of interest are fused to two different split reporter fragments; PPI induces recombination of the reporter and generation of an active reporter. Depending on the nature of the reporter, different signals can be detected, including genetic changes, cell death/survival and the products of enzymatic activities. Different enzymes have been used as scaffolds for the design of split reporters [85]: these include β-galactosidase, β-lactamase, ubiquitin, dihydrofolate reductase, thymidine kinase, TEV protease, horseradish peroxidase to cite a few. PCA can be compatible with different molecular imaging techniques such as electron microscopy (EM) or positron emission tomography (PET) [86]. Assays using fluorescence or luminescence as read-outs have been particularly useful in drug discovery as these signals can be easily quantified, especially in a high-throughput format. Recombination of a fluorescent protein [87] or luciferase and its derivatives [88] is used in bimolecular fluorescence complementation (BiFC) [89] or bimolecular luminescence complementation (BiLC) [90] assays. BiFC and BiLC have been used to detect PPIs in a variety of organisms, from yeast [91] to plants [92]. Both techniques have been instrumental in detecting and targeting membrane proteins, in particular G-protein coupled receptors (GPCRs) [93,94] as well as have successfully guided the identification of new antiviral [95] and anticancer [96] molecules. Many groups have explored the interactions of transcription factors with other proteins as well as with DNA using different versions of PCAs [97,98,99,100,101]. Recently the generation of large collections of vectors that allow performing large-scale PPI detection both in vitro and in living organisms has further increased the value of such techniques [102,103]. 

The main limiting factor for PCA development is the need to design a specific reporter for every protein pair under investigation. With PCA fragments fused directly to the protein of interest, the number of configurations to explore rapidly become time and resources-consuming [104]. A new strategy where anti-GFP or anti-Cherry nanobodies are fused to the split fragments of the small luciferase nanoLuc (nLuc) [105] could help bypassing this hurdle. This new system has been successfully used to detect PPI and protein oligomerization and can be extrapolated as a competitor/inhibitor screen. 

### 3.4. Functional Assays

Transcription is a complex process mediated by a network of proteins; its modulation often results in broad cellular variations whose phenotypic sum is difficult to predict. Functional validation of transcriptional therapy, at the level of the organism, is therefore ineluctable.

#### 3.4.1. RNA-seq, ChIP-seq, ChIP-MS

RNA-Seq takes advantage of the recent developments in next-generation sequencing (NGS) to reveal the presence and quantity of RNA in a biological sample, analysing the continuously changing cellular transcriptome. Understanding the transcriptome is key if we are to connect this information to TF function, overall GRN activity and ultimately, the impact of drugs on this network. 

Identifying protein complexes present on chromatin has been extremely challenging but recent advances in genomics and proteomics have delivered methods to interrogate these diverse and low frequency events. Proteins interacting with chromatin marks can directly be identified upon immunoprecipitation using a tagged protein, synthesized histone tails containing posttranslational modifications (PTMs) or various DNA baits. The combination of chromatin immunoprecipitation (ChIP) assays with sequencing, ChIP sequencing (ChIP-Seq) is a powerful method for identifying genome-wide DNA binding sites for transcription factors and other proteins [106]. In this assay, the DNA-bound protein is immunoprecipitated using a specific antibody. The bound DNA is coprecipitated, purified, and sequenced [107]. ChIP-Seq enables thorough examination of the interactions between proteins and nucleic acids on a genome-wide scale. 

Combining ChIP with mass spectrometry (ChIP-MS) allows the identification of GRNs in their in vivo context. ChIP-MS baits can be proteins in tagged or endogenous form, histone PTMs, or lncRNA and this variety opens up a plethora of potential for TF target ID. Unfortunately, the conventional ChIP methodology is not amenable to industrial scale-up and automation, due to the amount of hands-on time, total experiment time, and the prohibitively high quantity of sample and reagents required. 

Efforts to improve ChIP methodology have largely been successful in reducing sample and reagent requirements to thousands of cells per assay [108,109], but have not provided yet any scalable, automatable solutions. However, the birth of HTChip in 2012 opened the door to scaling up epigenetic screening with the creation of a high throughput, low consumption, and automated microfluidic device. Since then, a multitude of laboratories across the world, both in academia and industry are pinpointing the problems and automating those laborious aspects of the processes. Recently, Dainese et al. comprehensively addressed the limitations of standard ChIP-seq by developing a novel automated, microfluidic system named FloChIP. FloChIP is faster (<2 h), with more dynamic range (from 10^6^ to 500 cells), and higher-throughput capability (with the capacity of up to 64 parallel, antibody- or sample-multiplexed experiments run) compared to classical ChIP seq [110]. 

#### 3.4.2. Developmental Models

Model systems, including laboratory animals, microorganisms, and cell- and tissue-based systems, are central to the discovery and validation of drugs in complex human disease. The four most common animal models for genetic analysis are *Caenorhabditis elegans*, *Drosophila melanogaster*, *Danio rerio* (zebrafish), and *Mus musculus* (mice), chosen for their convenience. Forward or reverse genetic experiments conducted in model animals are used to identify the role of orthologous genes in humans, particularly genes relevant to human disease. Recent publications have begun focusing on the differences between human diseases and animal models of disease [111] and have noted in some cases the failure of the latter to predict therapeutic efficacy [112]. Because of this, some have advocated abandoning animal studies and focusing on clinical trials in human patients [113]; however, the fact remains that the ethical and monetary hurdles to primary screening of molecules in humans are insurmountable.

This led to the development of new approaches to validate cellular and animal models of disease and harmonise their behaviour with human disease. These advances include reverse translation of human monogenetic disease to establish homologous cell-based or animal disease models [114,115], the use of induced pluripotent stem cells [116], and molecular fingerprinting of diseased tissues in human versus animal models [117]. The current utilization of novel tissue and cell-based systems is beginning to allow a human focus from the start and, in the context of TF networks, will facilitate the comprehension of the role of tissue-specific TFs network in health and disease.

#### 3.4.3. In Silico Techniques

The high-throughput screening (HTS) of large proprietary compound collections and combinatorial libraries has increased the pressure on gathering pharmacokinetic and drug metabolism data as early as possible [118]. Progress has been made in in silico methods using various quantitative structure-activity relationship (QSAR) [119] to predict small molecule efficiency. Properties related to absorption, distribution, metabolism and excretion (ADME) can now predicted by algorithms (e-ADME) [120]. These in silico approaches are promising filters for virtual libraries [121]. 

## 4. Targeting Transcription Factors: Examples

As previously stated, although transcription was traditionally considered as undruggable, agents are now being developed that targets various levels of transcriptional regulation including DNA binding by transcription factors, protein-protein interactions, and epigenetic alterations. Here we review three promising examples, proof-of-concept for the *transcripto-therapy* of the future. 

### 4.1. p53

The p53 tumor suppressor is a principal mediator of growth arrest, senescence, and apoptosis in response to a broad array of cellular damage [122]. Rapid induction of high p53 protein levels by various stress types prevents inappropriate propagation of cells carrying damaged, potentially mutagenic, DNA. p53 can kill cells via a transcription-dependent function, in the nucleus and a transcription-independent function at the mitochondria [123]. Expression level of p53 has been shown to be the single most important determinant of its function. In normal unstressed cells, p53 is highly unstable with a half-life of 5 to 30 min. Therefore, very low cellular levels are present due to continuous degradation. Conversely, many cellular stress pathways such as DNA damage, hypoxia, telomere shortening, and oncogene activation induce the rapid stabilization of p53 via inhibition of its degradation. p53 degradation is largely mediated by MDM2 which, over the past decade, has emerged as the principal cellular antagonist of p53, limiting the p53 tumor suppressor function [124]. Interfering either with the interaction between p53 and MDM2, or with the ability of MDM2 to target p53 for degradation, leads to stabilization and activation of p53 in cells. 

Important efforts have led to the identification of compounds that inhibit p53-MDM2 binding. Several lead structures have been identified and optimized for potency and selectivity. One such class was a series of cis-imidazoline analogs called Nutlins (for Nutley inhibitor). These compounds displaced recombinant p53 protein from its complex with MDM2 with median inhibitory concentration (IC50) values in the 100 to 300 nM range [125].

The story of the nutlin discovery illustrates well the use of biophysical tools in drug development. Indeed, multiple techniques were instrumental to the discovery, as shown in Figure 4. For example, the inhibition of MDM2-p53 binding was analyzed with Biacore surface plasmon resonance technology in a solution-competition format. To investigate the mode of binding of these compounds, the crystal structure of the human MDM2–Nutlin-2 complex was obtained [125]. The resulting structure showed the inhibitor bound to the p53 binding site on MDM2, when compared to a previous structure of MDM2 bound to a 15-residue peptide from the transactivation domain of p53. For in vitro and in vivo monitoring of p53-MDM2 PPI, ELISA assays were used. Recently, a series of probes with a turn-on switch was also developed for monitoring p53–MDM2 interaction. These small molecule fluorescent probes are environment-sensitive turn-on fluorescent probes that have been successfully applied to imaging p53–MDM2 interaction in human lung cancer cell line. 

### 4.2. SOX18

SOX18 belongs to the SRY-related, HMG box (SOX) family of proteins. SOX genes are found throughout the animal kingdom and encode a highly conserved family of TFs involved in a wide range of developmental process such as lens formation, sex and neural determination, spermatogenesis, chondrogenesis, and cardiac development [126]. The pharmaceutical interest in SOX18 is due to its crucial role during the development of the cardio-vascular system. Recent transcriptomic results identified re-expression of SOX18 in neo-carcinoma [127], leading to the creation of new blood vessel around the tumors. This new vascularization is a condition *sine qua non* for the development of the tumor, as well as for the formation of metastasis. Therefore, therapeutic manipulation of neo-vascularization may provide an important adjunct treatment for tumor growth. So far only anti-VEGF signaling molecules have been developed to counteract tumor-induced de novo vessel formation as, for reasons we have previously discussed, TFs have not reached their full potential as molecular targets. 

As SOX18 TF functionality is dependent on or modulated by the presence of other co-binders, targeting of this GRN was an attractive anti-angiogenic strategy. The first step was to identify co-binders of SOX18 using ChIP-MS as a first line of screening [128]. These data have been coupled with a validation by AlphaScreen and single molecule fluorescence approaches that were optimised to scrutinise pairwise protein interactions in high throughput. This type of study design is ideal to rapidly identify high confidence PPIs in vitro and provided the field with 30 novel SOX18 interactions, including the discovery of the SOX18 homodimer [129]. This interactome analysis also provided a suite of SOX18 interactions that could be “drugged”.

After identifying potential drug targets, Overman et al. [128] screened a structurally diverse marine extract library for compounds that could interfere with SOX18-DNA interaction, using a fluorescence polarisation assay. From this screen, the authors identified the chemical space that was conserved between multiple compound hits. After screening for potential transcriptional off-target effects of the compounds through cell-based reporter assays, and in vitro and in vivo efficiency using cell lines and transgenic zebrafish reporter models, researchers chose one salicylic acid derivative, referred to as SM4, as the lead compound. To investigate the mode of action of the compounds, AlphaScreen was used again to study whether PPIs were affected. It showed that SM4 was capable of interfering with some of the SOX18 protein interactions, such as SOX18-SOX18, SOX18-DDX17 and SOX18-RBPJ. Interestingly, not all SOX18 PPIs were affected. This indicates that SOX18 likely has multiple interacting surfaces and the effect of SM4 is dependent on a particular surface of the protein. In addition, they found that chemical modifications to SM4 dramatically changed its PPI disrupting capacity, opening the door to fine-tuning of SOX18 activity [130]. In line with this, they reported that antibodies and antibody fragments could be used to modulate transcriptional output [131]. 

To assess drug efficacy for the functional inhibition of TFs, biological systems should be studied during either embryogenesis or pathogenesis. Therefore, in the case of SOX18, embryonic vascular development and corresponding tumour-induced angiogenesis needed to be studied as relevant contexts [128]. First, a zebrafish model was used to investigate treatment-induced vascular malformation as a phenotypic readout of small molecule activity and compared to SOX18 genetic depletion. Combining these approaches confirmed the effect of SM4 on SOX18-dependent embryonic process. Then, the pharmacological inhibition of SOX18 was investigated under pathological conditions, in a mouse model of breast cancer. SM4 was shown to reduce tumour vascularization and strongly reduce the metastatic spread, leading to increased survival. Together, this exemplifies the benefit of using a platform of in vitro target-based drug discovery followed by in vivo phenotype-based lead validation and optimization. 

### 4.3. Ets Family (ETS)

The Ets family (ETS) in mammals contains approximately 30 genes, all homologous to Ets-1, the first cellular homologue of the viral oncogene *v-ets* from the avian transforming retrovirus E26 [132]. The gene products of this family are TFs controlling various cellular functions in cooperation with other families of TFs and co-factors. All ETS possess a transcriptional activation or repression domain and an evolutionarily conserved Ets domain for DNA binding. In many human tumours, aberrant expression of apoptosis-related genes as well as growth-related genes like ETS has been observed [133]. Target genes for ETS transcriptional activity include oncogenes, tumour suppressors, apoptosis-related, differentiation-related, angiogenesis-related invasion and metastasis-related genes [134,135]. As such, ETS are involved not only in the malignant transformation of cells, but also in the promotion and progression of tumours by activating invasion and metastasis-related genes [136]. As a result, ETS are prime candidate molecular targets for cancer therapy. 

Ets-1 seems to be one of the most promising candidates, because targeting of this protein would possibly not only directly inhibit the proliferation and resistance to apoptosis of tumour cells, but also indirectly inhibit tumour growth and progression, including invasion and metastasis, through the inhibition of tumour angiogenesis [137]. The potential of repressing Ets1 activity has been shown experimentally. The first demonstration came through the use of dominant negative mutants. Introduction of a dominant-negative mutant of Ets-1 effectively inhibited growth of tumour cells in culture and neo-angiogenesis in vivo. Neo-angiogenesis induced by local inoculation of FGF in mouse ears was also significantly inhibited by expression of an Ets-1 mutant bearing a deletion of its activation domain [138]. The second validation was through the direct repression of ETS itself. Over-expression of Ets1/PEA3 into breast cancer cells expressing high levels of HER2/neu resulted in suppression of HER2 expression and prolonged survival with inhibited tumour growth in mice. In this case, a phase I clinical trial has just been completed in which HER-2/neu is downregulated [139]. The third and the fourth strategy using RNA interference [140] and antisense siRNA oligonucleotides [141] respectively are still under development. Despite the therapeutic potential, no small molecule inhibitors of Ets1 have been reported so far. Understanding the GRN of ETS transcription factors through studies such as those described previously will no doubt lead to a molecular targeting therapy against ETS, offering a novel approach to selective cancer treatment.

#### Clinical Trials for Transcription Therapy

These are still early days in the development of transcription therapy. Nevertheless, a handful of compounds are in clinical development and have made it to clinical trials (see selected examples in Table 1) [142,143,144,145]. The best example of TF-targeted drugs that entered clinical trials are the aforementioned p53-MDM2 PPI inhibitors, based on the nutlin scaffold or other chemical backbones. Roche has been the first company to push a p53-MDM2 inhibitor (RG7388) to trial and it has been followed by other compounds from companies such as Sanofi, Merck or Novartis to cite a few. A number of compounds directed at STAT3 or HIF1 are also under clinical evaluation. DNA-binding compounds such as trabectedin (ET-743) is approved for use for soft-tissue sarcoma under the orphan drug status in the US and is undergoing clinical trials for the treatment of other cancers. Although its mode of action is complex, one mode of action involves preventing the binding of FUS-CHOP to DNA and preventing its transcriptional activity.

Aberrant fusions between a transcription factor and an unrelated protein are causative of different pathologies and are therefore promising drug targets. This is the case of MLL oncofusions leading to leukemias. Small molecules have been developed to perturb PPIs of these oncofusions and reduce the downstream gene activation. Such compounds (MI-538 or MI-1481) have been approved by the US Food and Drug Administration as investigational new drugs and are entering phase I clinical trials. Another oncofusion between EWS and the transcription factor FLI1 from the ETS family is the most common cause of Erwing sarcoma. TK216, an inhibitor of PPI targeting the interaction between EWS-FLI and RNA helicase I, is currently undergoing phase I trial. Trabectedin has also been shown to target EWS-FLI and is under evaluation in the treatment of Ewing sarcoma. 

Despite the early successes of this type of approaches, there are several challenges in the structural design of PPI inhibitors. First, the inhibitor should bind relatively strongly to the protein target. This is achieved by mimicking the interactions between two proteins, especially those interactions between residues that contribute most to the binding energy, so-called binding hot spots. Second, many anti-cancer therapies are prone to acquired drug resistance which is a major challenge in cancer treatment. Therefore, the designed inhibitor (peptide or small molecule) should retain its properties, even if two interacting target proteins undergo extensive selection in the tumor to eliminate binding to these inhibitors while retaining binding between the two proteins. The latter task can be accomplished by inhibitor design along with in-silico mutagenesis. Finally, the inhibitors should be active with respect to not only the proposed PPI targets but also with respect to their paralogs. 

## 5. Summary and Perspectives

Drug development constantly adapts to the emergence of new technologies with wider applications. These emerging techniques can either provide new information (for example, conformational changes) or the same information (*k*on, *k*off, *K*d and stoichiometry) in a faster, simpler or more high-throughput manner. Many of the techniques we introduced use completely new types of physical principles for detection. While establishing new techniques, crucial factors should be kept in mind, in addition to the information they can provide. These include material requirements, ease of use and implementation in current drug discovery settings, as well as cost. To conclude, we will discuss emerging techniques that have the potential to revolutionize drug discovery and development pathways. 

### 5.1. Cryo-EM

Cryo-EM is establishing itself as a central tool in structural biology. The ability to obtain near-atomic resolution structures using electron microscopy was shown initially almost three decades ago in the context of electron crystallographic studies of membrane proteins [146]. Continued advances in single-particle Cryo-EM over the next two decades enabled high-resolution analysis of non-crystalline samples with high internal symmetry such as icosahedral and helical viruses [147]. Technological leaps in the design of sensors and use of algorithms has widen the use of cryo-EM and nowadays atomic resolutions of protein structures using cryo-EM are becoming more common [148] So far, all of the near-atomic-resolution structures reported have been of proteins with sizes in the range of ∼200 kDa or larger, and informal opinion in the field is that cryo-EM technology is primarily suited for analysis of relatively stable proteins with sizes > 150 kDa [149]. To date, the smallest protein for which a cryo-EM structure has been reported using single particle cryo-EM is that of the 93 kDa cancer target isocitrate dehydrogenase at 3.8 Å resolution [150]. The challenge remains in achieving near-atomic resolution for small proteins. Currently, the high level of effort involved in determining a structure limits the applications to non-routine, low-throughput and high-value projects, however further technological and analytical developments will overcome this obstacle.

### 5.2. Single-Molecule Imaging Techniques

Traditional biochemical, genetic, and genomic approaches have proved successful at identifying factors, regulatory sequences, and potential pathways that modulate transcription. However, these assays typically provide snapshots or population averages of the highly dynamic, stochastic biochemical processes involved in transcriptional regulation. Single molecule live-cell imaging has, therefore, emerged as a complementary approach capable of circumventing these limitations [151]. Recent advances in imaging techniques have enabled visualizations of nascent transcripts or individual protein molecules at high spatiotemporal resolution. Studies using single molecule fluorescent in situ hybridization (smFISH) [152] have revealed that only a fraction of cells in a population is transcriptionally active at any given time point. This heterogeneity in gene expression shows that transcription occurs in bursts rather than continuously [153], a behaviour that had been inferred from previous observations of many genes across various cell types and organisms. In the past three years, transcriptional bursting was directly observed by single molecule techniques for Nanog in mouse embryonic stem cells [154] and β-globin in erythroid cells [155]. Extensive studies have also been performed to understand the property of transcriptional bursting, such as bursting frequency, size, and durations. Many studies have tried to show the link between the regulation of bursting kinetics and the level of transcription factors. For example, using live imaging, Larson et al. controlled the level of steroids through light activation, and showed that steroids mediate the level of steroid-responsive genes by modulating bursting frequency [156].

### 5.3. Molecular Imaging

Molecular imaging allows non-invasive assessment of biological and biochemical processes in living subjects. Such technologies, therefore, have the potential to enhance our understanding of disease and drug activity during preclinical and clinical drug development. It could guide the selection of candidates that are most likely to be successful or to halt the development of drugs that seem likely to ultimately fail. Molecular imaging quantifies biological and molecular processes at the cellular and subcellular level in intact living subjects. Depending on the technique, it can require the use of specific molecular probes or rely on intrinsic tissue characteristics as the source of image contrast. Integrative biology, early detection and characterization of disease, as well as evaluation of treatment [157] could benefit. The advantage of molecular imaging techniques over more conventional readouts is that they can be performed in the intact organism with sufficient spatial and temporal resolution to study biological processes in vivo. Furthermore, these techniques allow for a repetitive, non-invasive, uniform and relatively automated study of the same living subject, thus, harnessing the statistical power of longitudinal studies. 

Two types of imaging can be distinguished: primarily morphological/anatomical and primarily molecular imaging techniques. Primarily morphological/anatomical imaging technologies, such as computed tomography (CT), magnetic resonance imaging (MRI) (with contrast agents injected at millimolar blood concentrations) and ultrasound have high spatial resolution but cannot identify diseases until the stage where tissue structural changes (for example, growth of a tumour) are large enough. Primarily molecular imaging modalities, such as optical imaging, positron-emission tomography (PET) and single-photon emission computerized tomography (SPECT) (with radiotracers injected at nanomolar blood concentrations) [158] can detect molecular and cellular changes of diseases before the tumour causes structural changes. However, so far, these imaging modalities have a poor spatial resolution. Combining the strengths of morphological/anatomical and molecular imaging modalities, using multimodality hardware and/or co-registration post-acquisition processing, allows the detection of pathophysiological changes in early disease phases at high structural resolution with techniques such as PET–CT or PET–MRI [159]. Other technologies such as high-resolution multidetector and dual-source CT technology with high temporal resolution and volumetric reconstruction capabilities, dynamic contrast enhanced MRI and CT, as well as magnetic resonance spectroscopy (MRS) are blurring the distinction between morphological and molecular imaging by also providing functional information [160,161]. Most large pharmaceutical companies have now established morphological imaging, as well as molecular imaging, as an integral part of both research and development, at various stages of the pipeline.

### 5.4. Personalized Medicine

As the scientific community was discovering the multitude of target genes contributing to human illness, genetic variability in patients’ responses has been recorded for various treatments. Consequently, scientists are now developing diagnostic tests based on genetics to better anticipate patients’ responses to targeted therapy [162]. However, several difficulties, both scientific and politic, must be overcome to achieve therapeutic efficacy. One of the main scientific challenges is for example determining which genetic markers have the most clinical significance. Policy challenges include finding a level of regulation for genetic tests that both protects patients and encourages innovation to identify genetic variants that are correlated with a drug response. 

The success of personalized medicine directly correlates with precise diagnostic tests that identify patients who can benefit from specific therapeutic approaches. For example, clinicians now commonly use diagnostics to determine which breast tumours overexpress the human epidermal growth factor receptor type 2 (HER2), in order to predict the response to the medication trastuzumab [163]. 

Recent studies are identifying genetic variations directly correlated with the risks of both rare and common diseases. These newly discovered genes, proteins, and pathways provide very powerful new drug targets, but on the other hand, there is insufficient indication of a downstream market to seduce the private sector to explore most of them. The current alternative to bridge this gap is to merge all the data set and for academics to become more translational. The NIH and the FDA are developing a more integrated pathway that connects all the steps between the identification of a potential therapeutic target by academic researchers and the approval of a therapy for clinical use. This path includes NIH-supported centres where scientists can screen thousands of chemicals to find potential drug candidates, as well as public–private partnerships to help move candidate compounds into the next step of commercial development. To conclude, real progress will come with a change in mentality: one drug may not fit all, and the clinical response needs to be stratified based on the patients’ genetics. These new types of data set will reverse the way to think about therapy and will offer a better strategy for compounds development.

## 6. Conclusions

In this review we demonstrate that biophysical technologies, which detect molecules physically interacting with one another, are now employed in drug discovery and especially in transcription therapy. This is because targeting new therapeutic candidates such as transcription factors is different from designing classical enzyme inhibitors and requires moving beyond traditional biochemical assays that detect enzyme/inhibitor activity. The thorough understanding of GRNs through the identification of novel co-binders will allow the detection of inhibitors of these protein complexes (PPI inhibitors) and ultimately the development of fragment-based drugs and molecular therapies. The implementation of novels techniques alongside a mentality shift towards personalized medicine will pave the way for an exciting and promising era for transcription therapy.

## Figures and Tables

**Figure 1 ijms-21-02301-f001:**
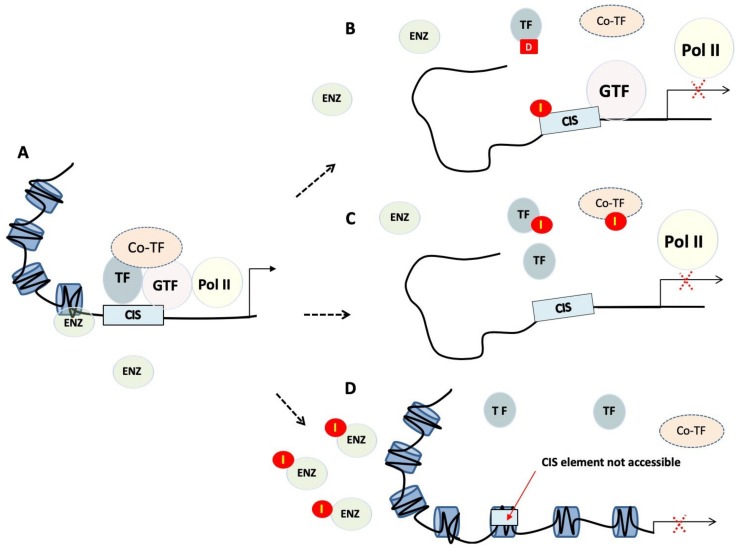
Transcriptional regulation and targeting strategies. (**A**) Transcriptional regulation is the means through which a cell regulates the conversion of DNA to RNA and so thereby orchestrates gene activity. RNA polymerases (Pol II), transcription factors (TF), as well as a multitude of other proteins act in concert to regulate this activity. (**B**) Small molecules or polyamides (I) compete with transcription factors binding to cis-regulatory elements, whereas decoys (D) bind transcription factors preventing them from binding to promoters. (**C**) Peptide mimetics or small molecules disrupt dimerisation of transcription factors, or interactions between transcription factors and their co-regulators. (**D**) Tight or closed chromatin is more compact and so refractory to factors that need to gain access to the DNA template. TF, transcription factor; GTF, general transcription factor; Pol II, RNA polymerase II; Co-TF, transcription co-regulator; I, inhibitor; D, transcription factor decoy; ENZ, modifying enzymes.

**Figure 2 ijms-21-02301-f002:**
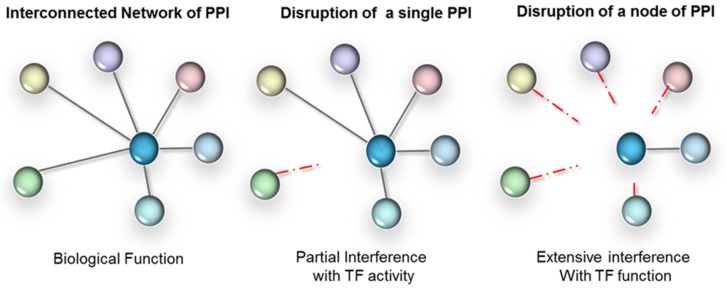
PPI interference strategy for TFs. Left panel: TFs form extensive networks of PPI, have many gene targets and regulate complex biological processes. Parallel signalling often overlaps functionally. Middle panel: interference with a single PPI (mutant or drug based) will only affect those target genes and functions that rely on those interactions. Right panel: a strategy that relies on broad-scale interference with a node of interactions is likely to more completely ‘knock-out’ the TF functionally.

**Figure 3 ijms-21-02301-f003:**
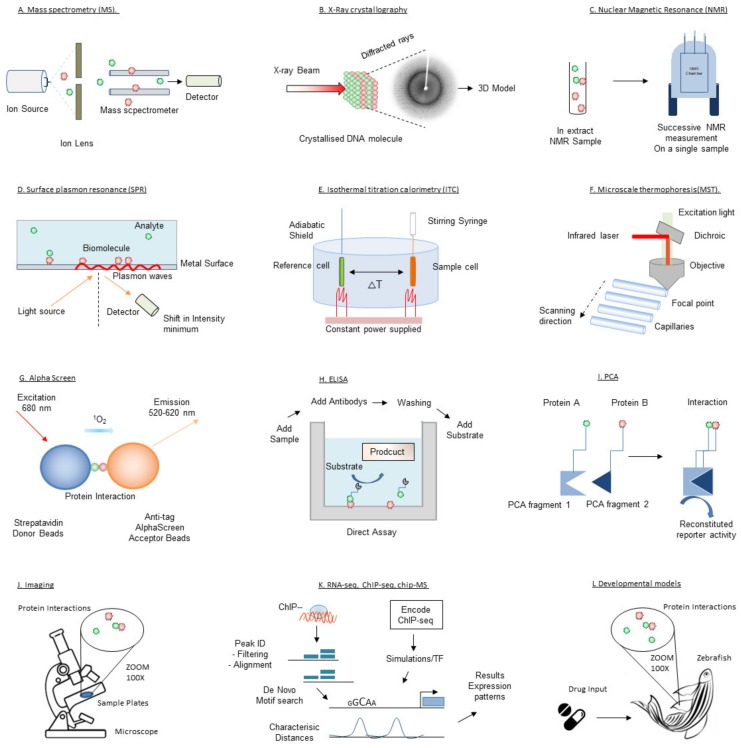
Schematic depiction of the techniques introduced in this review.

**Figure 4 ijms-21-02301-f004:**
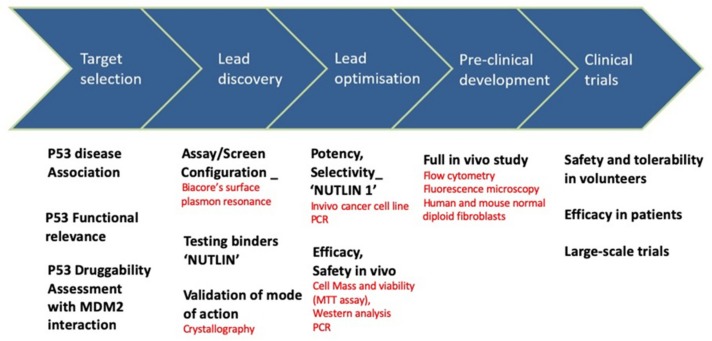
Nutlin: drug discovery and development and associated techniques.

**Table 1 ijms-21-02301-t001:** Selected examples of drugs undergoing clinical trials for future transcription therapy. Respective targets and studies (National Clinical Trial (NCT) Identifier Numbers) are indicated.

Drug	Target	National Clinical Trial (NCT)
		NCT02407080
RG7388/idasanutlin	p53/MDM3	NCT03287245
		NCT02828930
		NCT03362723
		NCT03107780
AMG232	p53/MDM2	NCT02016729
		NCT01723020
		NCT02110355
		NCT00955812
OPB-31122	STAT3	NCT00511082
		NCT01406574
		NCT00657176
		NCT01423903
OPB-51602	STAT3	NCT01344876
		NCT01184807
		NCT01867073
ET743	DNA	NCT01692678 NCT01343277
		NCT00070109
		NCT01453283
BC- 2059/ Tegavivint	TBLl/CTNNBl	NCT03459469
E-7386	CREB/CTNNBl	NCT03833700
		NCT03264664
		NCT01711034
OPB-111077	STAT3 SH2 interactor	NCT03197714
		NCT03158324
MK6482	HIFl complex	NCT04195750
TK216	EWS-FLI/RHA	NCT02657005

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
