# Peer review of "Biophysical Techniques for Target Validation and Drug Discovery in Transcription-Targeted Therapy"

_ijms, 2020, doi:10.3390/ijms21072301_

Round 1
Reviewer 1 Report
In this review, Moustaqil et al review biophysical techniques associated with target validation with a special focus on transcriptionally targeted molecules. Overall, this review accurately summarizes the state of the field and is largely inclusive of modern technologies. This manuscript will make a suitable contribution to IJMS; however, I firmly suggest that the senior authors on this manuscript edit the language throughout for readability and logical sentence construction.
Issues:
- The English throughout is poorly used and would benefit from a thorough review. Many sentences are difficult to understand.
- The title is not exactly reflective of the content within. Nowhere in the manuscript are target identification techniques (presumably from small molecule phenotypic screens) discussed in detail. A more accurate title would be “biophysical techniques for target validation and drug discovery in transcription targeted therapy.”
- Figures throughout are in different typefaces. Ideally this would be a sans-serif font for all.
- Figure 2 is difficult to read due to small font size.
- Typos in Figure 2: A should be “mass spectrometer”; H should be “antibodies”, G should be “streptavidin”; K should be “Characteristic distances”
- Typo in Figure 4: should read “in vivo” under lead optimization panel
Author Response
We thank the reviewers for their comments and addressed the different issues.
Reviewer 1 :
In this review, Moustaqil et al review biophysical techniques associated with target validation with a special focus on transcriptionally targeted molecules. Overall, this review accurately summarizes the state of the field and is largely inclusive of modern technologies. This manuscript will make a suitable contribution to IJMS; however, I firmly suggest that the senior authors on this manuscript edit the language throughout for readability and logical sentence construction.
We apologize for this, the manuscript has been entirely re-read and edited to ensure readability (see red annotations).
Issues:
The English throughout is poorly used and would benefit from a thorough review. Many sentences are difficult to understand.
See above.
The title is not exactly reflective of the content within. Nowhere in the manuscript are target identification techniques (presumably from small molecule phenotypic screens) discussed in detail. A more accurate title would be “biophysical techniques for target validation and drug discovery in transcription targeted therapy.”
The title has been amended as suggested: “Biophysical techniques for target validation and drug discovery in Transcription targeted therapy”.
Figures throughout are in different typefaces. Ideally this would be a sans-serif font for all.
This has been changed; all figures are now in Calibri sans serif, 11 pt.
Figure 2 is difficult to read due to small font size.
Title texts in Figure 3 have been increased to 14 pt. Description text are now in 12 pt font instead of 10 pt.
Typos in Figure 2: A should be “mass spectrometer”; H should be “antibodies”, G should be “streptavidin”; K should be “Characteristic distances”
These typos have been corrected.
Typo in Figure 4: should read “in vivo” under lead optimization panel
These typos have been corrected.

Reviewer 2 Report
This is a nice and succinct overview on a very complex topic, namely biophysical techniques for target identification and drug discovery in transcription therapy. The authors should consider the following points:
1. In the Protein-protein interactions interrogation techniques, such as Affinity chromatography, immunoprecipitation, Förster resonance energy transfer, In silico techniques. The biotechniques should be listed and the limitation and advantages of each technique should be discussed.
2. in the 2.4. Functional Assays, the imaging analysis to detect the output of TF modulation in the cell phenotypes should be discussed.
3,. The examples of the successful or clinical trials of PPI durgs and their taget should be discussed.
Author Response
We thank the reviewers for their comments and addressed the different issues.
Reviewer 2:
This is a nice and succinct overview on a very complex topic, namely biophysical techniques for target identification and drug discovery in transcription therapy. The authors should consider the following points:
- In the Protein-protein interactions interrogation techniques, such as Affinity chromatography, immunoprecipitation, Förster resonance energy transfer, In silico techniques. The biotechniques should be listed and the limitation and advantages of each technique should be discussed.
We added short paragraphs about affinity chromatography, immunoprecipitation, Förster resonance energy transfer, in silico techniques in the text.
- in the 2.4. Functional Assays, the imaging analysis to detect the output of TF modulation in the cell phenotypes should be discussed.
We added a new paragraph about single molecule imaging techniques development.
- The examples of the successful or clinical trials of PPI durgs and their taget should be discussed.
New paragraphs as well as a new table (Table 1) have been added.
